# The Effect of Hypoxia on the Expression of CXC Chemokines and CXC Chemokine Receptors—A Review of Literature

**DOI:** 10.3390/ijms22020843

**Published:** 2021-01-15

**Authors:** Jan Korbecki, Klaudyna Kojder, Patrycja Kapczuk, Patrycja Kupnicka, Barbara Gawrońska-Szklarz, Izabela Gutowska, Dariusz Chlubek, Irena Baranowska-Bosiacka

**Affiliations:** 1Department of Biochemistry and Medical Chemistry, Pomeranian Medical University in Szczecin, Powstańców Wielkopolskich 72 Av., 70-111 Szczecin, Poland; jan.korbecki@onet.eu (J.K.); patrycja.kapczuk@pum.edu.pl (P.K.); patrycjakupnicka@o2.pl (P.K.); dchlubek@pum.edu.pl (D.C.); 2Department of Anaesthesiology and Intensive Care, Pomeranian Medical University in Szczecin, Unii Lubelskiej 1, 71-281 Szczecin, Poland; klaudynakojder@gmail.com; 3Department of Pharmacokinetics and Therapeutic Drug Monitoring, Pomeranian Medical University in Szczecin, Powstańców Wielkopolskich 72 Av., 70-111 Szczecin, Poland; barbara.gawronska-szklarz@pum.edu.pl; 4Department of Medical Chemistry, Pomeranian Medical University in Szczecin, Powstańców Wlkp. 72 Av., 70-111 Szczecin, Poland; izagut@poczta.onet.pl

**Keywords:** CXC chemokine, cancer, tumor, hypoxia, hypoxia-inducible factor, cycling hypoxia, HIF-1α, NF-κB, SDF-1, IL-8

## Abstract

Hypoxia is an integral component of the tumor microenvironment. Either as chronic or cycling hypoxia, it exerts a similar effect on cancer processes by activating hypoxia-inducible factor-1 (HIF-1) and nuclear factor (NF-κB), with cycling hypoxia showing a stronger proinflammatory influence. One of the systems affected by hypoxia is the CXC chemokine system. This paper reviews all available information on hypoxia-induced changes in the expression of all CXC chemokines (CXCL1, CXCL2, CXCL3, CXCL4, CXCL5, CXCL6, CXCL7, CXCL8 (IL-8), CXCL9, CXCL10, CXCL11, CXCL12 (SDF-1), CXCL13, CXCL14, CXCL15, CXCL16, CXCL17) as well as CXC chemokine receptors—CXCR1, CXCR2, CXCR3, CXCR4, CXCR5, CXCR6, CXCR7 and CXCR8. First, we present basic information on the effect of these chemoattractant cytokines on cancer processes. We then discuss the effect of hypoxia-induced changes on CXC chemokine expression on the angiogenesis, lymphangiogenesis and recruitment of various cells to the tumor niche, including myeloid-derived suppressor cells (MDSCs), tumor-associated macrophages (TAMs), tumor-associated neutrophils (TANs), regulatory T cells (T_regs_) and tumor-infiltrating lymphocytes (TILs). Finally, the review summarizes data on the use of drugs targeting the CXC chemokine system in cancer therapies.

## 1. Introduction

Growing standards of living and advances in medicine in many countries have resulted in increased life expectancies. This has led to the spread of other diseases, particularly “civilization diseases” which are associated with lifestyle and environmental pollution, and diseases related to old age. One of these is cancer, a major problem in developed countries [1] and the cause of 55.9 million deaths worldwide in 2017 [1,2]—a 25% increase when compared to 2007 [2]. This trend has led to the intensification of efforts aimed at developing more effective therapeutic approaches.

The last 15 years have seen an increasing interest in the tumor microenvironment—a set of factors affecting a cancer cell in a tumor [3]. One of these are noncancer cells recruited to a tumor niche, with the recruitment process depending on factors secreted from cells already located in the tumor niche. These include factors changing the recruited cell into a tumor-associated cell—i.e., transforming the cell into an element which supports cancer processes [4,5]. Both cancer and noncancer cells secrete a number of factors which shape the tumor microenvironment, including chemoattractant cytokines, also known as chemokines [6,7]. Changes in the expression of chemokines affect the recruitment of cells to the tumor niche, angiogenesis and migration of cancer cells. For this reason, understanding the role of chemokines in cancer processes seems crucial for understanding the functioning of a tumor [7]. There is a lack of research on the influence of hypoxia, i.e., tumor-specific conditions, on CXC chemokines and so the aim of this review was to collect all literature data on the influence of hypoxia on the expression of CXC chemokines in a tumor.

## 2. Hypoxia in a Tumor

Tumors are associated with two kinds of hypoxia—chronic hypoxia (also known as continuous or noninterrupted hypoxia) and cycling hypoxia (intermittent or transient hypoxia). Chronic hypoxia is a persistent reduction in oxygen levels that results from an excessive distance between cells and vascular vessels whose growth cannot keep up with the rapid growth of the tumor [8]. Cycling hypoxia has cyclic episodes of oxygen deficiency interspersed with periods of reoxygenation [9]—a result of structural abnormalities in the vascular vessels, particularly the lack of a conventional hierarchy (Figure 1) [10,11]. In such vessels, the blood flow often changes route, resulting in hypoxia in some areas of the tumor. Then, when the blood flow changes its route again, some areas become reoxygenated. One such cycle of hypoxia lasts from several minutes [12] to 4 h [13], depending on the selected research model.

Both types of hypoxia change the expression of similar genes [14] as they activate the same transcription factors—hypoxia-inducible factors (HIFs) and nuclear factor κB (NF-κB). The mechanisms of activation, however, are different. In chronic hypoxia, the main role is played by a reduction in oxygen levels which triggers a drop in the activity of oxygen-dependent enzymes [15]. In cycling hypoxia, transcription factors are activated mainly by reactive oxygen species (ROS) [16,17].

Hypoxia-inducible factors (HIF-1, HIF-2 and HIF-3) exhibit transcriptive activity that is regulated by the proteolytic degradation of subunit α (HIF-1α and HIF-2α) [18,19,20]. This degradation occurs due to the hydroxylation of HIF-1α and HIF-2α in proline residues by prolyl hydroxylase (PHD), which leads to the ubiquitination by von Hippel-Lindau protein (pVHL) and then to proteolytic degradation of these HIF-α subunits [21]. PHD are oxygen-dependent enzymes (Figure 2) [15]. For this reason, in chronic hypoxia, there is a decrease in PHD activity which leads to a decrease in HIF-α hydroxylation of proline residues and a decrease in proteolytic degradation of these HIF subunits. HIF-1α reaches the maximum level in a cell after 6 h of chronic hypoxia, whereas HIF-2α reaches this after 48 h [22]. The expressions of PHD2 and PHD3 are increased by HIF-1 and HIF-2 [23,24,25,26], which makes HIF-α degrade more intensely during reoxygenation. In turn, cycling hypoxia increases the level of ROS, which oxidizes the iron atoms in PHD and thus reduces the activity of these enzymes [16,27]. This leads to a drop in the hydroxylation of proline residues on HIF-α, which means that HIF-α is no longer degraded but accumulated instead. The increase in HIF-1α levels during cycling hypoxia also occurs via other pathways—for example, the ROS-dependent activation of protein kinase A (PKA) phosphorizes this HIF-1 subunit thereby increasing its stability [28,29]. A ROS-dependent increase in calcium ion concentration in cytoplasm, the activation of protein kinase C (PKC) and the mammalian target of rapamycin (mTOR) are also significant [30]. In cycling hypoxia, the accumulation of HIF-1α increases with the successive cycles of oxygen deficiency [31]. However, this protein is completely degraded during episodes of reoxygenation [31].

The regulation of HIF transcriptional activity also involves factor inhibiting HIF (FIH), an enzyme which causes the hydroxylation of asparagine residue [32] and thus disrupts the interaction of HIF-α with the CBP/p300 coactivator [33,34]. In this way, FIH blocks the expression of HIF/CBP/p300-dependent genes. As FIH is an oxygen-dependent enzyme [35], its activity is reduced by chronic hypoxia and ROS which oxidize the iron atom—a part of the enzyme that is critical for its function [32]. FIH requires a much lower level of oxygen for its activity than PHD (K_m_ for FIH is 90 μM, whereas for PHD it is 230–250 μM) [15,35,36]. For this reason, in moderate hypoxia, the HIF pathway is inhibited by FIH but not by PHD.

Both types of hypoxia also activate NF-κB. In chronic hypoxia, this activation is dependent, among other things, on a reduction in the amount of oxygen and the ensuing reduction in the activity of PHD, an enzyme that inhibits the activation of NF-κB and HIF [37,38]. The activation of NF-κB in chronic hypoxia also depends on the activation of calcium/calmodulin-dependent kinase 2 (CaMK2) [39]. In contrast, in cycling hypoxia, NF-κB activation is ROS-dependent [17,40,41].

In chronic hypoxia, the activation of NF-κB results in an increased expression of HIF-1α and therefore has a significant effect on the HIF activation pathway [42,43]. This is related to the occurrence of the NF-κB binding site in the *HIF1A* gene promoter. For this reason, during chronic hypoxia, the expression of genes is directly induced by NF-κB. There is also an increase in the expression of genes directly dependent on HIF-1α but these are also indirectly dependent on NF-κB during chronic hypoxia. Therefore, in order to acquire a detailed insight into the mechanism which induces the expression of a given gene by chronic hypoxia, it is necessary to demonstrate the occurrence and investigate the functionality of the hypoxia responsive element (HRE) binding HIF or the NF-κB binding site. Importantly, chronic hypoxia and inflammation exclude each other by various mechanisms [44,45], and so chronic hypoxia reduces the inflammatory response. On the other hand, some proinflammatory genes are induced by both chronic hypoxia and inflammation [46]. Cycling hypoxia is more proinflammatory than chronic hypoxia [41,47,48]. This is related to the activation of NF-κB by ROS [17,40,41]. For this reason, NF-κB plays a more important role in gene expression during cycling hypoxia than in chronic hypoxia.

Hypoxia significantly changes the functioning of the tumor. Its proapoptotic effect on cells results in a selection of cells in terms of apoptosis resistance, a process which is important at the beginning of tumor development and results in the presence of cancer cells with a p53 dysfunction in the tumor [49]. Hypoxia also participates in the progression of cancer at further stages of the process. In particular, hypoxia is important in the functioning of cancer stem cells (CSCs) [50,51,52,53,54,55], which increase the resistance of the tumor to anticancer therapy. Hypoxia also causes cancer cell migration, invasion and metastasis, partly due to hypoxia causing the epithelial-to-mesenchymal transition (EMT) [55,56,57,58,59]. For this reason, areas of chronic hypoxia are often associated with neoplastic cell metastasis.

Tissues respond to oxygen deficiency by developing new blood vessels. In this way, hypoxia increases the expression of proangiogenic factors such as vascular endothelial growth factor (VEGF)-A [60,61], platelet-derived growth factor subunit A (PDGF-A), transforming growth factor-β (TGF-β) and angiopoietin-like 4 (ANGPTL4) [62]. Hypoxia also participates in tumor immune evasion by polarizing macrophages to the M2 phenotype which silences the immune response [63]. It also protects cancer cells by impairing the function of NK cells [64,65] and increasing the production of immunosuppressive proteins such as indoleamine 2,3-dioxygenase (IDO), human leukocyte antigen-G (HLA-G), programmed death-ligand 1 (PD-L1) and metabolites such as adenosine [66,67]. The hypoxia-induced acidosis of the cancer microenvironment, which is caused by an increased production and secretion of lactate is also important [66,68]. Lactate causes tumor immune evasion and neoplastic cell migration.

Hypoxia also affects the CXC chemokine system, which leads to changes in the level of these chemoattractant cytokines in the cancer microenvironment. CXC chemokines participate in the growth of the tumor due to a number of procancer properties.

HIF-1α accumulation and increased HIF-1 transcriptional activity occurs in cancer cells even in normoxia. This is related to, among other things, mutations in the *VHL* gene which encodes pVHL, resulting in the loss of biological function of pVHL, thereby reducing the degradation of HIF-1α [69,70]. Tumors also exhibit deletions of parts of the chromosome where the *HIF1AN* gene locus are located [71]. This gene encodes FIH-1, the enzyme responsible for inhibiting the transcriptional activity of HIF-1. Another way of activating the HIF-1 pathway under normoxia is HIF-1α phosphorylation [20] which leads to the increased stability of this protein and consequently, to HIF-1α accumulation in cells and increased expression of HIF-1dependent genes. Enzymes performing such phosphorylation under normoxia include PKA activated by cAMP [72], phosphatidylinositol-4,5-bisphosphate 3-kinase (PI3K) [73,74], extracellular signal-regulated kinase (ERK) mitogen-activated protein kinase (MAPK) [75,76,77]. HIF-1α is also phosphorylated by glycogen synthase kinase 3 which reduces the stability of this protein [77]. Apart from phosphorylation, other types of post-translational modifications also affect the activation of HIF-1 under normoxia. One of them is the deacetylation of HIF-1α by sirtuin 1 (SIRT1), which leads to a decrease in HIF-1 transcriptional activity [78]. In addition to the post-translational modification, the activation of NF-κB in normoxia also increases the expression of HIF-1α, which is important for an increase in the activation of HIF-1 during inflammatory reactions [75]. The elevated activation of HIF-1 via these pathways in normoxia increases the expression of genes dependent on this transcriptional factor. Although there are no studies in this area that confirm it, it is possible that this mechanism is responsible for the increase in the expression of CXC chemokines and CXC chemokine receptors in normoxia.

## 3. CXC Chemokines in Cancer

CXC chemokines (also known as α-chemokines) are a subfamily of chemotactic cytokines with a CXC motif at the N-terminus. Currently, we know 17 CXC chemokines [6,79], of which CXC motif chemokine ligand (CXCL)15 is a murine CXC chemokine not found in humans [80]. As such, it is not mentioned in Table 1 which shows the procancer and anticancer properties of CXC chemokines. The chemokines from this subfamily activate the CXC motif chemokine receptors (CXCRs)1–8 (Table 1) [6,79,81,82,83,84]. The receptor for CXCL14 is not clearly established. It seems that CXCL14 is a CXCR4 antagonist [85]. However, subsequent studies have shown that CXCL14 may not directly affect CXCR4 [86]. Other studies have shown that the receptors for CXCL14 may be G-protein coupled receptor 85 (GPR85) [87], toll-like receptor (TLR)9 after the formation of a complex of CXCL14 with CpG oligodeoxynucleotide [88], and insulin-like growth factor I receptor (IGF-IR) [89].

CXC chemokines differ in their effect on angiogenesis. Seven of them (CXCL1, CXCL2, CXCL3, CXCL5, CXCL6, CXCL7, and CXCL8), with an additional ELR motif, mainly induce angiogenesis via CXCR2 [90,91,92]. VEGF activity also induces an increase in the expression of CXCL1 and CXCL8 in endothelial cells which enhances angiogenesis [93,94]. In contrast, CXCL4, CXCL9, CXCL10 and CXCL11 exert an angiostatic effect via their CXCR3 receptor [95,96,97]. One of them, CXCL4, has been shown to inhibit lymphangiogenesis, and so it can be inferred that the remaining CXCR3 ligands may have the same properties [97]. Another CXC chemokine, CXCL12, has angiogenic properties [98], and its actions are associated with VEGF, which increases the expression of CXCR4 (receptor of CXCL12) on endothelial cells, which in turn enhances the influence of CXCL12 on these cells [99]. CXCL12 induces an increase in VEGF expression in endothelial cells and thus indirectly enhances angiogenesis [100]. The activation of CXCR4 on a tumor cell also causes an increase in VEGF-C expression, leading to lymphangiogenesis [101]. Of the remaining CXC chemokines, CXCL13 interferes with the action of basic fibroblast growth factor (bFGF) which inhibits angiogenesis [102], CXCL14 also has angiostatic properties [103,104] and CXCL16 is an angiogenic chemokine [105], similar to CXCL17, a chemoattractant for monocytes and macrophages in which it increases the expression of VEGF-A [106].

CXC chemokines can cause a migration and invasion of tumor cells [7,107,108,109]. CXCR3 ligands either induce or inhibit migration of tumor cells, depending on whether CXCR3-A or CXCR3-B is activated [110,111]. CXCL14, depending on the research model, can have anticancer properties [112] and procancer properties [113].

Akin to other chemokines, CXC chemokines participate in the recruitment of cells to the tumor niche but what makes them different is their distinct ability to recruit tumor-associated neutrophils (TANs) [114,115,116,117,118,119,120,121]. Apart from TAN, they recruit myeloid-derived suppressor cells (MDSCs) [122,123,124,125,126,127], tumor-associated macrophages (TAMs) [122,128,129,130], mesenchymal stem cells (MSCs) [131,132,133,134,135] and regulatory T cells (T_regs_) [136,137,138,139,140,141,142,143,144,145,146]. CXCL14 stimulates the autocrine growth of cancer-associated fibroblasts (CAFs) [147]. On the other hand, the ligands of CXCR3, and CXCL14 and CXCL16, cause tumor infiltration by anticancer tumor-infiltrating lymphocytes (TILs) [148,149,150,151,152,153] and thus show anticancer properties.

## 4. Ligands of Receptors CXCR1 and CXCR2

### 4.1. The Effect of Hypoxia on the Expressions of CXCL1 and CXCL2

The effect of hypoxia on the expressions of CXCL1 and CXCL2 depends on the type of cancer. For example, chronic hypoxia does not affect the expression of either chemokine in lung adenocarcinoma [154] but does increase the expression of CXCL1 in human acute myeloid leukemia cells [155], cervical cancer cells [156] and hepatocarcinoma cells [157]. The expressions of both CXCL1 and CXCL2 are increased by chronic hypoxia in PC-3 prostate cancer cells [14] but reduced in SK-OV-3 ovarian adenocarcinoma and WM793B melanoma cells. In colitis-associated colon cancer, the source of CXCL1 in hypoxia may be tumor epithelial cells which, in a reaction mediated by HIF-2, increase the expression of this chemokine in the tumor microenvironment resulting in the recruitment of monocytes to the tumor niche [158].

Cycling hypoxia increases the expression of both described chemokines (CXCL1 and CXCL2) in PC-3 prostate cancer cells and SK-OV-3 ovarian adenocarcinoma cells [14]. In contrast, it reduces the expression of CXCL1 and CXCL2 in WM793B melanoma cells [14]. However, in an in vivo model of Lewis lung carcinoma, cycling hypoxia increases the level of CXCL1 in a tumor [47].

Another important source of CXCL1 and CXCL2 in the hypoxic microenvironment may be TAMs and MDSCs, as chronic hypoxia increases CXCL1 and CXCL2 expression in macrophages [159] and MDSCs [160]. The most important factor increasing the expression of CXCL2 in TAMs is HIF-2 [161]. HIF-1-dependent expression of both chemokines in MDSCs is also significant [160]. This effect on the hypoxia-induced production of CXCL1 and CXCL2 in TAMs and MDSCs may be associated with the presence of HRE in the promoters of the *CXCL1* and *CXCL2* genes [158,162,163].

### 4.2. Effect of Hypoxia on the Expression of CXCL3

The promoter of the *CXCL3* gene does not contain HRE [162]. For this reason, chronic hypoxia does not affect the expression of CXCL3 in lung adenocarcinoma cells [154] and WM793B melanoma cells [14]. Nevertheless, chronic hypoxia does increase the expression of CXCL3 in primary human monocytes [164], PC-3 prostate cancer cells [14] and MDSCs [160]. In MDSCs, this effect is dependent on HIF-1 [160], although this mechanism requires further research. HIF-1 may indirectly affect the expression of CXCL3. It can increase the expression of a protein or miRNA which directly regulates the expression of this chemokine. In contrast, chronic hypoxia reduces the expression of CXCL3 in primary rat astrocytes [165] and SK-OV-3 ovarian adenocarcinoma cells [14].

Cycling hypoxia also influences the expression of CXCL3. Its effect also depends on the cell line, inducing a reduction in expression in WM793B melanoma cells and an increase in PC-3 prostate cancer cells [14]. It may also have no effect at all, as in SK-OV-3 ovarian adenocarcinoma cells [14].

### 4.3. Effect of Hypoxia on CXCL5 Expression

Chronic hypoxia does not increase the expression of CXCL5 in breast cancer cells [166], cervical squamous carcinoma cells [167], lung adenocarcinoma cells [154] or MDSCs [160]. In contrast, chronic hypoxia increases the expression of this chemokine in primary human monocytes [164]. As well as this, the overexpression of HIF-2α in murine articular chondrocytes increases the expression of CXCL5 which is related to the presence of HRE in the *CXCL5* gene promoter [163]. The incompatibility of these results with those mentioned at the beginning of the paragraph may be due to differences between species. It is also possible that the expression of CXCL5 is dependent on HIF-2 but not on HIF-1. The indicated studies analyzed the changes in cells after 24 h of hypoxia—in such a long period of hypoxia, HIF-1 is activated, but HIF-2 is not. The maximum activation of HIF-2 only occurs after 48 h of chronic hypoxia [22].

### 4.4. Effect of Hypoxia on CXCL6 Expression

Chronic hypoxia increases the expression of CXCL6 in hepatocellular carcinoma cells [168], which induces the migration of these cancer cells. Chronic hypoxia also increases the expression of CXCL6 in PC-3 prostate cancer cells [14]. The induction of CXCL6 expression is HIF-1-dependent because the *CXCL6* promoter contains HRE [162,168]. However, chronic hypoxia does not always enhance the expression of CXCL6 as it can also reduce it in primary human monocytes [164] and HeLa cervical cancer cells [169]. Chronic hypoxia has no effect on the expression of this chemokine in breast cancer cells [166], the HepG2 hepatocarcinoma cell line [157] and lung adenocarcinoma cells [154].

In turn, cycling hypoxia increases the expression of CXCL6 in PC-3 prostate cancer cells [14].

### 4.5. Effect of Hypoxia on CXCL7 Expression

Chronic hypoxia increases CXCL7 expression in cervical cancer cells [169] and primary rat astrocytes [165]. The overexpression of HIF-2α increases the expression of this chemokine in murine articular chondrocytes [163]. In contrast, chronic hypoxia does not alter CXCL7 expression in HepG2 hepatocellular carcinoma cells [157] and lung adenocarcinoma cells [154].

### 4.6. Effect of Hypoxia on CXCL8 Expression

A tumor contains areas which are distant from blood vessels and characterized by chronic hypoxia, acidosis and cell necrosis. These areas show increased expression of CXCL8 as demonstrated by in vivo studies in human melanoma [170,171], human pancreatic cancer [172] and ovarian carcinoma [173]. The observed effect depends on hypoxia [172,174,175] and the acidosis of this microenvironment [172,176]. Chronic hypoxia does not alter CXCL8 expression in some cells, such as breast cancer cells [177], lung adenocarcinoma cells [154], primary hepatocytes [178] and uveal melanoma cells [179].

An increase in CXCL8 expression in melanoma cells [180], ovarian carcinoma [173,181], pancreatic cancer [182] and rhabdomyosarcoma [183] in chronic hypoxia is dependent on the activation of activating protein-1 (AP-1) and NF-κB—in particular, NF-κB p65, NF-κB p50 and c-Jun [173]. The increase in CXCL8 expression in these tumor cells is independent of HIF-1 [173,183] and nuclear factor-interleukin-6 (NF-IL-6) [173].

As with all the aforementioned chemokines, the mechanism of an increase in CXCL8 expression in chronic hypoxia depends on the type of tumor. In glioblastoma multiforme cells [184], an increase in CXCL8 expression in chronic hypoxia is AP-1-dependent but not NF-κB-dependent. In contrast, in some bladder cancer lines, it is associated with the activation of NF-κB but not AP-1 [185]. In turn, in hepatocellular carcinoma cells, an increase in the expression of CXCL8 in chronic hypoxia is dependent on the activation of HIF-1 and the effect of this transcription factor on NF-κB [186,187]. In other types of cancer, the effect of HIF-1 may differ—for example, in colon cancer cells, chronic hypoxia increases the expression of CXCL8 via NF-κB [188]. However, the activation of HIF-1 inhibits the effect of NF-κB on the expression of CXCL8 [188].

Hypoxia can also indirectly increase the expression of CXCL8 in glioblastoma multiforme cells by a HIF-1-dependent increase in the expression of ZNF395, a transcriptional factor that supports the increase in CXCL8 expression [189]. In melanoma cells, chronic hypoxia degrades Tip110 by increasing the ubiquitination of this protein [190]. The decrease in the level of Tip110 causes an increase in CXCL8 mRNA stability, which leads to an increase in CXCL8 expression [191]. In colon cancer, chronic hypoxia intensifies the expression of dual specificity phosphatase 2 (DUSP2), a nuclear phosphatase which reduces the activation of MAPK cascades [192,193]. The reduction in the expression of DUSP2 increases the activation of MAPK cascades in chronic hypoxia and thus elevates the expression of CXCL8. In turn, in lung adenocarcinoma cells, chronic hypoxia causes a change in the substrate specificity of dihydrodiol dehydrogenase which then starts to produce prostaglandin F_2α_ (PGF_2α_), which increases the expression of CXCL8 [194]. However, it needs to be remembered that the mechanisms mentioned in this paragraph are not necessary for the induction of CXCL8 expression in tumor cells by chronic hypoxia and that they only enhance the influence of chronic hypoxia in terms of the expression of this chemokine.

Cycling hypoxia also increases the expression of CXCL8, although its effect is dependent on the research model. In PC-3 prostate cancer cells, both types of hypoxia induce a similar increase in the expression of CXCL8 [14]. In SK-OV-3 ovarian adenocarcinoma cells, cycling hypoxia—unlike chronic hypoxia—increases the expression of CXCL8 [14]. In human aortic endothelial cells, cycling hypoxia increases the expression of CXCL8 more than chronic hypoxia [195]. In neutrophils, cycling hypoxia increases the expression of CXCL8 and the receptor of this chemokine CXCR2, which reduces the spontaneous apoptosis of these cells [196,197]. This may be significant in the recruitment and accumulation of TANs into the tumor niche. The effect of cycling hypoxia on CXCL8 expression is associated with increased ROS levels and greater NF-κB activation than in chronic hypoxia [198,199].

TAMs may also be an important source of CXCL8 in the hypoxic microenvironment. However, there are no studies showing in vivo effects of hypoxia on CXCL8 expression in TAMs. The expression of this chemokine in TAMs may be increased by factors from tumor cells [200]. Studies on monocyte-derived macrophages have shown that the expression of this chemokine in these cells may be increased by chronic hypoxia [159,201] via AP-1 activation [201]. HIF-1 and HIF-2 are no less important in this process [159], influencing the activation of NF-κB and thus increasing the expression of CXCL8. However, studies on (i) phorbol 12-myristate 13-acetate (PMA)–differentiated THP-1 macrophages and (ii) PMA–differentiated THP-1 macrophages, polarized by lipopolysaccharides (LPSs) and interferon-γ (IFN-γ), have shown that chronic hypoxia does not affect the expression of CXCL8 in these cells [202]; however, cycling hypoxia increases the expression of this chemokine in these THP-1 macrophages, in a process dependent on the activation of c-Jun N-terminal kinase (JNK) MAPK and NF-κB [202].

The expression of CXCL8 in endothelial cells is an important issue in cancer. VEGF secreted by cells in the tumor niche causes an increase in CXCL1 and CXCL8 expressions in endothelial cells, in a process important to angiogenesis and the migration of cancer cells [93,94]. Chronic hypoxia changes the expression of CXCL8 in endothelial cells, although this effect is dependent on the research model. For example, an increase in CXCL8 expression has been observed in human umbilical vein endothelial cells (HUVECs) [203], human pulmonary microvascular endothelial cells and human pulmonary aortic endothelial cells [204]. This effect depends on the activation of the p38 MAPK and PI3K→PKB/Akt pathway, which in turn activates AP-1, HIF-1 and NF-κB [204]. The activation of HIF-2 also plays a significant role in this process [205].

In human microvascular endothelial HMEC-1 cells, hypoxia reduces CXCL8 expression (Figure 3) [206,207] in a process mediated by HIF-1 and HIF-, which regulate the expression of CXCL8 [208]. HIF-1 increases the expression of Bach1, which in turn reduces the expression of erythroid 2-related factor (Nrf2) [207]. This reduces the expression of CXCL8. HIF-1 also decreases the expression of c-Myc. HIF-2 has the opposite effect as it increases the expression of c-Myc [208]. HIF-2 also stabilizes the c-Myc:Max complex which increases c-Myc action, unlike HIF-1 which destabilizes the complex. As c-Myc increases the expression of CXCL8 by binding to the CXCL8 promoter, both HIFs can indirectly alter the expression of CXCL8, with a more pronounced influence of HIF-1.

Cycling hypoxia alone does not increase the expression of CXCL8 and does not affect the activation of NF-κB, as shown in HUVECs [47]. Nevertheless, this type of hypoxia does increase the response of these cells to proinflammatory factors such as tumor necrosis factor α (TNF-α). This results in a more intense activation of NF-κB and expression of CXCL8 and ICAM-1 in HUVECs than would occur as a result of TNF-α alone, which in turn increases the adhesion of these cells to the walls of blood vessels and thus their recruitment into the tumor niche.

### 4.7. Influence of Hypoxia on CXCR1 and CXCR2 Expression

CXCR2 is a receptor of the CXCL1, CXCL2, CXCL3, CXCL5, CXCL6, CXCL7 and CXCL8 chemokines, and CXCR1 is receptor of CXCL6 and CXCL8 [6,79,81]. Changes in the expression of these receptors affect the action of these chemokines. Chronic hypoxia increases the expression of CXCR1 and CXCR2 in cervical carcinoma cells [209] and prostate cancer cells [210]. This effect is dependent on HIF-1 and NF-κB activation [210]. Chronic hypoxia also increases the expression of CXCR2 on aortic endothelial cells [211]. This may indicate the effect of hypoxia on angiogenesis by increasing the sensitivity of endothelial cells to CXCR2 ligands. However, this effect is dependent on the type of tumor—for example, chronic hypoxia reduces the expression of CXCR2 in gastric cancer cells [212].

There are no data on the influence of cycling hypoxia on the expression of the described receptors on cancer cells. Screening studies did not show any influence on three cell lines: PC-3 (prostate cancer cells), SK-OV-3 (ovarian adenocarcinoma cells) and WM793B (melanoma cells) [14]. However, cycling hypoxia increases NF-κB-dependent expression of CXCR2 in neutrophil cells [197] but decreases CXCR1 expression [213]. Increasing the expression of CXCR2 increases the effect of CXCL1, CXCL2 and CXCL8 chemokines on these cells. In particular, there is a decrease in spontaneous apoptosis of these cells which supports TAN accumulation in the tumor niche [196,197].

## 5. Ligands of Receptor CXCR3

### 5.1. The Influence of Hypoxia on CXCL4 Expression

Chronic hypoxia increases the expression of CXCL4 in MUM2B uveal melanoma cells [214]. This effect is mediated by HIF-1 [214] and dependent on the type of tumor. For example, chronic hypoxia does not change the expression of CXCL4 in lung adenocarcinoma cells [154].

### 5.2. Effects of Hypoxia on CXCL9 Expression

Chronic hypoxia does not affect the expression of CXCL9 in cervical cancer cells [167] and lung adenocarcinoma cells [154]. Nevertheless, chronic hypoxia increases the expression of this chemokine in primary rat brain astrocytes [165] and hepatocarcinoma cells [157].

### 5.3. Effects of Hypoxia on the Expression of CXCL10

The *CXCL10* gene promoter contains one binding site for HIF and four binding sites for NF-κB [215]. For this reason, the overexpression of HIF-2 increases the expression of CXCL10 in murine articular chondrocytes [163]. Chronic hypoxia also increases the expression of CXCL10 in HepG2 cells (hepatocellular carcinoma) [216]. On the other hand, studies on cancer cells such as HeLa cells (cervical cancer) [156], 9L glioma cells (brain tumor) [217], breast cancer cells [175,177], ovarian cancer cells, colon cancer cells and lung cancer cells [175] show that chronic hypoxia reduces the expression of CXCL10, as in studies on noncancer cells such as primary hepatocytes [178] and rabbit alveolar macrophages [201]. This effect may depend on HIF-1 or on the reduction in transcriptional NF-κB activity [218]. However, in some types of cancer cells, such as acute myeloid leukaemia cells [155], chronic hypoxia does not change the expression of CXCL10.

There are also four binding sites for NF-κB in the *CXCL10* gene promoter [215]. For this reason, by activating NF-κB and without the involvement of HIF, chronic hypoxia increases CXCL10 expression in cardiac microvascular endothelial cells [215]. In the same way, cycling hypoxia increases the expression of CXCL10 in M1 polarized macrophages [202], although not in M2 macrophages [202].

### 5.4. Effect of Hypoxia on CXCL11 Expression

The influence of chronic hypoxia on the expression of CXCL11 depends on the research model. In HeLa cervical cancer cells [169] and primary human monocytes [164], chronic hypoxia reduces the expression of this chemokine. In cervical cancer, C33a cells increase [167]. However, chronic hypoxia does not affect the expression of CXCL11 in hepatocarcinoma cells [157].

### 5.5. Effects of Hypoxia on CXCR3 Expression

CoCl_2_ increases CXCR3 receptor expression in the model of renal cell carcinoma 786-O cells [219]. CoCl_2_ is a substance that mimics the effects of hypoxia [220]. For this reason, it can be assumed that chronic hypoxia increases the expression of CXCR3 on cancer cells.

## 6. Effect of Hypoxia on the Expression and Function of CXCL12 and CXCR4 and CXCR7 Receptors

### 6.1. Effect of Hypoxia on CXCL12 Expression

Chronic hypoxia affects the expression of CXCL12. However, this effect depends on the adopted research model. In WM793B melanoma cells [14] and GOT1 cells (ileal carcinoids) [221], chronic hypoxia reduces CXCL12 expression. Chronic hypoxia also reduces the expression of CXCL12 in MDSCs [222], which is related to the HIF-1-dependent increase in the level of miRNA-210. Chronic hypoxia does not affect the expression of CXC12 in breast cancer cells [166], gastric cancer cells [212], lung adenocarcinoma cells [154], rhabdomyosarcoma cells [183] and uveal melanoma cells [179]. However, it does increase the expression of CXC12 in cervical cancer cells [167], clear cell-renal cell carcinomas cells [223], glioblastoma multiforme cells [224,225,226], multiple myeloma plasma cells [227], ovarian carcinomas cells [228] and endothelial cells [229,230]. This effect is dependent on HIF-1 [212,223,224,227,231] and HIF-2 [158,227]. Transforming growth factor β (TGF-β) may also participate in the mechanism of increasing CXCL12 expression [224,232]. On the other hand, in HUVECs, the increase in CXCL12 expression is dependent on NF-κB p65, which increases the expression of long intergenic nonprotein coding RNA 1693 (LINC01693), which functions as a miRNA-302d sponge [230]. miRNA-302d reduces the expression of CXCL12—i.e., hypoxia increases the expression of this chemokine via NF-κB.

Cycling hypoxia, akin to chronic hypoxia, reduces the expression of CXCL12 in WM793B melanoma cells [14].

### 6.2. Effect of Hypoxia on the Expression of CXCL12 Receptors: CXCR4 and CXCR7

Chronic hypoxia increases the expression of CXCR4 in tumor cells such as breast cancer cells (Figure 4) [233,234,235,236,237,238], chondrosarcoma cells [239,240], clear cell-renal cell carcinomas cells [223], chronic lymphocytic leukemia [241], colon cancer cells [62,242,243], gastric cancer cells [212,244], glioblastoma multiforme cells [226,245,246,247], GOT1 cells (ileal carcinoids) [221], lung cancer cells [248,249], melanoma cells [250,251], multiple myeloma cells [252], oral squamous cell carcinoma cells [253], osteosarcoma cells [254,255], pancreatic ductal adenocarcinoma cells [256,257], renal cell carcinoma cells [258], and retinoblastoma cells [259]. However, chronic hypoxia does not universally induce CXCR4 expression in tumor cells—for example, in hepatocellular carcinoma cells it only increases the expression of CXCR4 in some cell lines [260].

The expression of CXCR7 under the influence of chronic hypoxia is upregulated in glioblastoma multiforme cells [261]. However, its expression is not altered by chronic hypoxia in acute myeloid leukemia cells [262] and breast cancer cells [237,243].

Cycling hypoxia, unlike chronic hypoxia, does not affect CXCR4 expression in PC-3 prostate cancer cells and SK-OV-3 ovarian adenocarcinoma cells [14]. However, cycling hypoxia increases CXCR7 expression in PC-3 prostate cancer cells, although this effect is not as intense as in chronic hypoxia [14]. In contrast to chronic hypoxia, cycling hypoxia does not affect the expression of CXCR7 in SK-OV-3 ovarian adenocarcinoma cells [14].

Expression of CXCR4 is increased directly by HIF-1 [62,159,212,223,229,239,241,244,248,252,253,254,255,258,263,264] and HIF-2 [159,161,229,248,265], because the promoter of the *CXCR4* gene contains HRE [263,266]. In particular, chronic hypoxia by HIF-2 increases the expression of CXCR4 in macrophages [161] and endothelial progenitor cells [267]. In breast cancer cells, chronic hypoxia increases CXCR4 expression independently of HIF-1, depending only on NF-κB [233]. Chronic hypoxia increases the expression of CXCR4 in the pancreatic ductal adenocarcinoma cell line by reducing the level of miRNA-150 [257]. In chondrosarcoma, chronic hypoxia increases the expression of miRNA-181a, which controls the expression of the regulator of G-protein signaling 16 (RGS16) [240,268]. This protein reduces the expression of CXCR4. As such, miRNA-181a increases the expression of CXCR4. In breast cancer cells CoCl_2_ (chemical hypoxia-mimicking agent) reduces miRNA-302a expression [238]. This miRNA regulates the expression of CXCR4—therefore, reducing the level of miRNA-302a increases the expression of CXCR4.

Hypoxia may also alter the level of CXCR4 expression in leukemias. In multiple myeloma cells, chronic hypoxia increases CXCR4 expression in a process dependent on HIF-1 [252]. However, hypoxia can interfere with the action of CXCL12 on these cells by HIF-2-dependent upregulation of CC motif chemokine receptor 1 (CCR1) [269]. Activation of this receptor reduces the action of CXCL12 on multiple myeloma cells. In leukemic monocytic cells, mild and severe chronic hypoxias reduce the expression of CXCR4 [270]. This is related to a HIF-1- and HIF-2-dependent increase in the level of miRNA-146a, which reduces the expression of CXCR4 in these cells. Chronic hypoxia also causes a decreased migration of leukemic cells in response to CXCL12. This is related to the disturbance of Akt/PKB activation as shown by experiments on the HL-60 promyelocytic leukemia cell line [234]. In acute myeloid leukemia, the dysfunction of CXCL12 by chronic hypoxia is dependent on cholesterol depletion, which interferes with the function of lipid rafts and thus the internalization of CXCR4 [271]. In contrast, in myeloma cell lines U266 and RPMI8226, chronic hypoxia increases the expression of CXCR4 by increasing the expression of miRNA-210 [272].

### 6.3. Consequences of Hypoxia’s Influence on the CXCL12→CXCR4 Axis

Hypoxia acts on the CXCL12→CXCR4 axis. However, the ramifications of this effect are too complex to be outlined in the section on tumor function of CXC chemokines. For this reason, consequences of the action of hypoxia on this axis are described in this part of the review.

The CXCL12→CXCR4 pathway induced by chronic hypoxia has a significant procancer effect, especially due to the angiogenic influence of CXCL12 [98]. In a tumor cell, chronic hypoxia increases the expression of VEGF and in some types of tumor cells it increases the expression of CXCL12 [228]. CXCL12 causes an increase in VEGF expression on endothelial cells [100]. VEGF increases expression of CXCR4 on HUVECs [99] and human brain microvascular endothelial cells [245]—sensitizing these cells to angiogenic CXCL12 [229]. Hypoxia also acts directly on endothelial cells by increasing the expression of VEGF [273] and CXCR4; the latter is shown by studies on HUVECs [230,259,263,274] and microvascular endothelial cells [250]. To a lesser extent, chronic hypoxia also increases the expression of CXCR4 on microvascular endothelial cells [250] and CXCL12 in HUVECs [229,230]. Importantly, it should be remembered that chronic hypoxia does not only result in angiogenesis by inducing the expression of CXCL12 and VEGF, but via other factors as well.

At the same time, a chronic hypoxia-induced increase in CXCR4 expression stimulates the proliferation of cancer cells [256,275] and causes their migration and invasion [244,246,248,249,255,256,257,275], especially in combination with the chronic hypoxia-induced increase in CXCL12 expression in tumor cells [167,223,228]. The upregulation of CXCR4 also aids in the transendothelial migration of cancer cells [229], which is followed by the invasion of cancer cells to neighboring tissues and organs [253,276,277,278]. In breast cancer, the CXCL12→CXCR4 axis also participates in another stage of metastasis. After reaching the bone, breast cancer cells are retained close to osteoprogenitor cells in the hypoxic niche [279]; the elevated expression of CXCL12 in the latter promotes the growth of breast cancer cells and thereby results in bone metastasis.

In many types of cancer cells, chronic hypoxia does not increase the expression of CXCL12 [154,166,179,183,212], although with an increased expression of CXCR4, the sensitivity of tumor cells to CXCL12 increases. In addition, cancer cell migration involving the CXCL12→CXCR4 axis may be caused by other cells—for example, CAFs in gastric cancer, where chronic hypoxia increases the expression of CXCL12 and thus acts on tumor cells with a hypoxia-enhanced expression of CXCR4 [212].

Chronic hypoxia also increases the expression of CXCR4 in monocytes [164,263] and macrophages [159,263]. For this reason, it can cause the accumulation of TAMs and MDSCs at sites with a high expression of CXCL12 [122], particularly in tumors [122,223,224,228]. A similar mechanism is postulated for the accumulation of T_regs_ in the hypoxic niche [136], and the recruitment of neural stem cells in glioblastoma multiforme [225] and CD34^+^ endothelial progenitor cells in pituitary adenomas [280].

Chronic hypoxia increases the expression of CXCR4 on multiple myelomas in a HIF-1-dependent manner [252]. This leads to the migration of these cells and their homing to bone marrow via the CXCL12→CXCR4 pathway [281]. Bones require physiological hypoxia for normal hematopoiesis [282,283], but are also involved in multiple myeloma. In this cancer, hypoxia increases the expression of angiogenic factors such as VEGF, CXCL8 and CXCL12 [283,284]. The upregulation of CXCL12 is induced by HIF-2 and to a lesser extent by HIF-1 [227], resulting in angiogenesis and osteoclastic bone resorption [227]. This, in turn, leads to angiogenesis and interference with the normal bone marrow microenvironment. Hypoxia in bone marrow can also cause the egress of multiple myeloma cells [269], as HIF-2 increases the expression of CCR1. Activation of this receptor by CC motif chemokine ligand 3 (CCL3) reduces the response of multiple myelomas to CXCL12 and the consequent egress of these cells from bone marrow.

CXCR4 also has a function inside the tumor cell. Its activation by CXCL12 triggers its entry and internalization into the nucleus, where it interacts with HIF-1, enhancing the expression of genes dependent on this transcription factor [285].

## 7. Influence of Hypoxia on the CXCL13→CXCR5 Axis

Chronic hypoxia does not alter CXCL13 expression in breast cancer cells [166], hepatocarcinoma cells [157] and lung adenocarcinoma cells [154]. Although the promoter of the *CXCL13* gene contains HRE [162], no data are available on the direct chronic hypoxia-induced upregulation of CXCL13 in a tumor cell. It is only known that chronic hypoxia increases the expression of CXCL13 in adipocytes [286]. On the other hand, chronic hypoxia may affect the expression of CXCL13 in the tumor indirectly—for example, via an increased expression of TGF-β in myofibroblasts in prostate tumors [287].

There are no reports of hypoxia affecting the expression of CXCR5 (CXCL13 receptor) in cancer cells. Nevertheless, the promoter of the *CXCR5* gene contains HRE [286]. For this reason, chronic hypoxia increases the expression of CXCR5 as demonstrated in experiments on adipocytes [286].

## 8. Effect of Hypoxia on CXCL14 Expression

Since the promoter of the *CXCL14* gene contains HRE [288], chronic hypoxia can increase the expression of CXCL14, for example, in cervical cancer cells [169]. The overexpression of HIF-2α in murine articular chondrocytes also increases the expression of CXCL14 [163]. Nevertheless, chronic hypoxia does not change the expression of the described chemokine in lung adenocarcinoma cells [154].

## 9. Effect of Hypoxia on CXCL15 Expression

CXCL15 is a murine chemokine that is not found in humans and for this reason has been poorly researched. CXCL15 is mainly expressed in murine lungs [289], and to a much lesser extent in the murine digestive tract and urogenital organs [80]. The expression of this chemokine can be increased by chronic hypoxia, as shown by experiments on murine articular chondrocytes, where the overexpression of HIF-2α resulted in an elevated expression of CXCL15 [163].

## 10. Influence of Hypoxia on the CXCL16→CXCR6 Axis

The effect of chronic hypoxia on the expression of CXCL16 depends on the type of cancer. In hepatocarcinoma cells [157] and HUVECs [105], chronic hypoxia increases the expression of CXCL16. In HUVECs, this effect is dependent on HIF-1 [105]. In breast cancer cells, chronic hypoxia also increases the expression of CXCL16, although this is dependent on the CXCL10–CXCL16 loop [290]. In contrast, chronic hypoxia does not alter CXCL16 expression in lung adenocarcinoma cells [154]. Finally, it increases the expression of the CXCL16 receptor (CXCR6) in breast cancer cells in a HIF-1-dependent manner [290,291].

## 11. Influence of Hypoxia on the CXCL17→CXCR8 Axis

There are no data available on the effect of hypoxia on the expression of CXCL17, or its receptor (CXCR8), in tumor cells. However, experiments on myocytes have shown that chronic hypoxia increases the expression of CXCR8 in a HIF-1-dependent process [292,293]. This indicates that hypoxia increases the expression of this receptor on cancer cells and thus enhances the effect of CXCL17 on these cells.

CXCL17 is a ligand of CXCR8 [294], although there is currently a discussion on the existence of another, yet undiscovered, receptor [295,296]. A potential discovery of a new receptor could reveal new methods of regulating the action of CXCL17, including the influence of hypoxia on the action of this chemokine.

## 12. Cancer Therapy

### 12.1. Cancer Therapy Targeting Hypoxia

Hypoxia and the signaling pathways activated by hypoxia are convenient therapeutic targets in cancer therapy (Figure 5). This is related to the fact that hypoxia occurs mainly in the tumor. For this reason, research focuses on inhibitors of the transcriptional activities of HIF-1 and HIF-2 [297]. On the other hand, oxygen concentration may be increased in the tumor and thus hypoxia may be prevented by using hyperbaric oxygen therapy [298,299]. However, as hypoxia is also important in the functioning of hematopoietic stem cells under physiological conditions [300,301,302], any therapy against hypoxia may also cause side effects which affect the production of various blood cells.

In another therapeutic approach, anticancer therapy targets certain proteins induced by hypoxia. A monoclonal anti-VEGF-A antibody known as bevacizumab is already used in therapy as an angiostatic factor [303,304]. Inactivation of PD-L1 is also being studied as a way to increase the effectiveness of anticancer immunotherapy [305,306]. However, targeting CXC chemokines remains a very important issue in cancer therapy, especially as hypoxia alters the expression of these chemoattractant cytokines, which promotes the growth of tumors.

### 12.2. Cancer Therapy Targeting CXC Chemokines

Changes in the expression of CXC chemokines under the influence of hypoxia are an important element of cancer processes. For this reason, CXC chemokines and their receptors are a convenient therapeutic target in the treatment of cancer, with most clinical and preclinical studies focusing on CXCR2 and CXCR4 receptors and their ligands.

The chemokines which act through CXCR1 and CXCR2 participate in numerous cancer processes [7], and therefore CXCR1/2 inhibitors [307,308,309] as well as CXCR2 inhibitors [310], are considered as potential anticancer drugs. They reduce the proliferation and migration of cancer cells and inhibit angiogenesis, and their combination with current anticancer drugs has shown promising results [307]. Anti-CXCL8 antibody [311,312], CXCL8 small interfering RNA [313], and anti-CXCL1 antibody [314] have also been tested and such therapies have shown a reduction in tumor cell migration and angiogenesis. Another therapeutic approach involves the transduction of CXCR2 to antitumor lymphocytes which accumulate in the tumor niche and exhibit anticancer activity there [315,316].

Drugs targeting the CXCL12→CXCR4 axis have also been investigated. Although the use of anti-CXCR4 antibodies are being tested in anticancer monotherapy [317], better results are obtained when drugs targeting this axis are combined with those already used in anticancer therapy. This approach involves CXCR4 antagonists, CXCR4 inhibitors, anti-CXCR4 antibodies and CXCR7 inhibitors [318,319,320,321,322]. One of the advantages of these drugs is their ability to decrease drug resistance—e.g., to tamoxifen by CXCR4 antagonists in breast cancer [323]. They also have the potential to increase the effectiveness of immunotherapy [324,325]. An anti-CXCR4 antibody, or peptides recognizing CXCR4, have also been tested to target cytotoxic substances against cancer stem cells in acute myelogenous leukemia [326], breast cancer cells [327], colorectal cancer cells [328] and multiple myeloma [329]. This method can also be used to retarget an adenovirus vector against a tumor cell in gene therapy [330].

Another direction of research is the use of gene therapy to increase the expression of chemokines that induce the infiltration of a tumor by anticancer TILs—for example, CXCR3 ligands such as CXCL10 [331,332,333] and CXCL11 [334]. Promising results are also shown in the combination of current therapeutic methods, e.g., radiotherapy, with gene therapy which increases the expression of CXCL10 [335]. However, CXCL10 induces tamoxifen resistance in breast cancer [336] and therefore any interference with the expression of the aforementioned chemokine during therapy should be performed after careful examination of its interaction with other anticancer drugs.

Drugs that interfere with the action of other CXC chemokine receptors are also being tested, although these studies are not as advanced. An example of this is drugs which block the CXCL13→CXCR5 axis, with in vitro studies on breast cancer cells showing that anti-CXCL13 antibodies induce tumor cell apoptosis [337]. However, to date, no advanced animal model studies or clinical trials have been conducted in this area.

Another promising chemokine is CXCL16, which causes tumor infiltration by anticancer TILs [153]. The expression and release of this chemokine in a tumor cell is increased by ionizing radiation [338], and so combining radiotherapy (increasing the expression of chemokines that result in the recruitment of anticancer lymphocytes) with immunotherapy (which consists of introducing modified anticancer lymphocytes into the patient’s body) has been postulated.

## Figures and Tables

**Figure 1 ijms-22-00843-f001:**
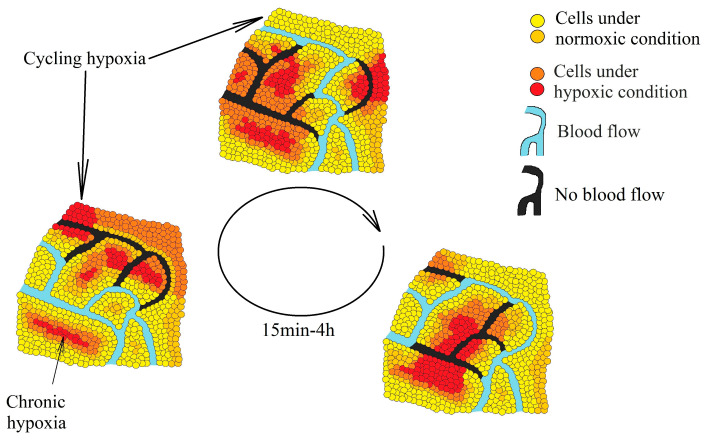
Two types of hypoxia in a tumor: chronic hypoxia associated with an excessive distance from the blood vessels and cycling hypoxia associated with changes in the blood flow in the vessels inside the tumor which lead to periodic oxygen deficiencies in various parts of the tumor.

**Figure 2 ijms-22-00843-f002:**
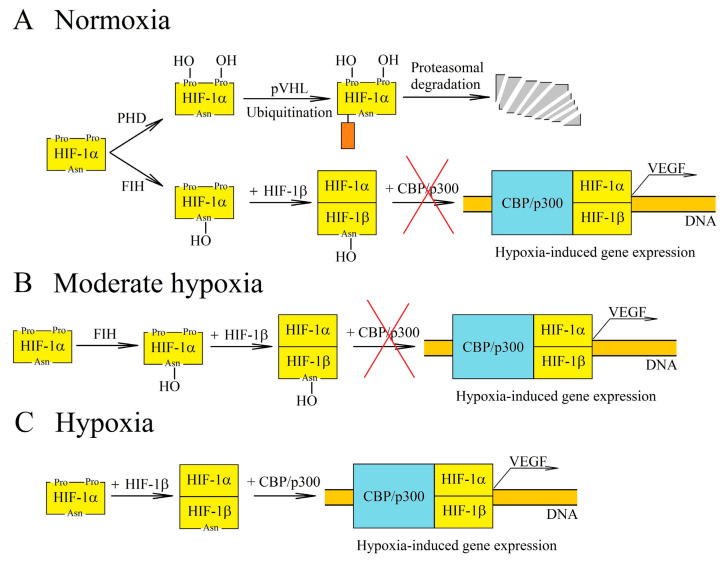
Regulation of hypoxia-inducible factor-1 (HIF-1) transcriptional activity at various oxygen concentrations. (**A**) In normoxia, HIF-1α is hydroxylated by the oxygen-dependent enzymes prolyl hydroxylase (PHD) and factor inhibiting HIF (FIH). The PHD-catalyzed reaction leads to proteasomal degradation of HIF-1α. In turn, hydroxylation by FIH prevents interaction of HIF-1 with the CBP/p300 coactivator. (**B**) PHD and FIH require different concentrations of oxygen for their activity and so, in moderate hypoxia, PHD activity decreases while FIH retains its functions. Accumulation of HIF-1α occurs, but due to hydroxylation by FIH, there is no interaction between HIF-1 and the CBP/p300 coactivator, and thus genes dependent on this coactivator are not expressed. (**C**) In hypoxia, the activities of oxygen-dependent enzymes are reduced. For this reason, HIF-1α is not hydroxylated by PHD or by FIH. This subunit begins to form a transcription factor with HIF-1β, which is responsible for inducing the transcription of hypoxia-dependent genes.

**Figure 3 ijms-22-00843-f003:**
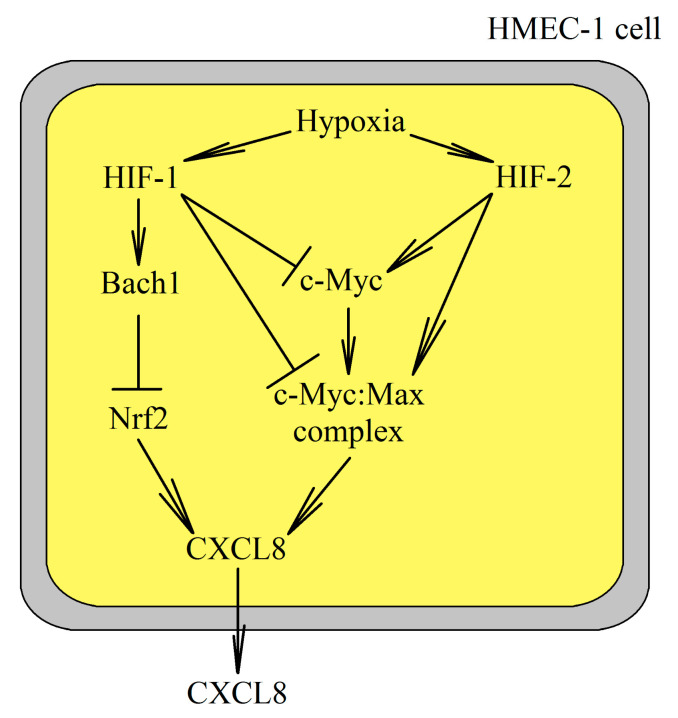
The mechanism of interaction between HIF-1 and HIF-2 in the change of CXCL8 expression in HMEC-1 cells. HIF-1 reduces CXCL8 expression in HMEC-1 cells due to an increase in Bach1 expression, which reduces erythroid 2-related factor (Nrf2) activation. HIF-1 also reduces the expression of c-Myc and destabilizes the c-Myc:Max complex. HIF-2 increases the expression of CXCL8 in these cells, which is related to an increase in c-Myc expression and stabilization of the c-Myc:Max complex.

**Figure 4 ijms-22-00843-f004:**
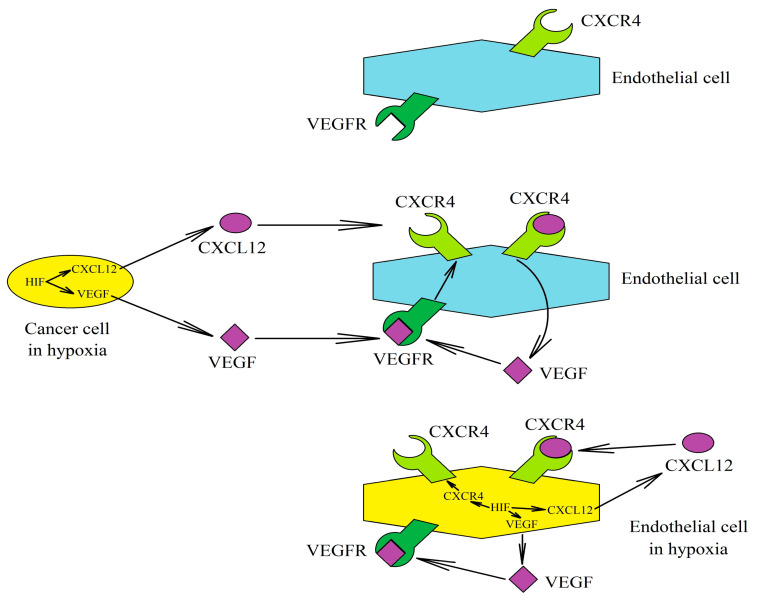
Association between the CXCL12→CXCR4 axis and vascular endothelial growth factor (VEGF) in angiogenesis induced by chronic hypoxia. Chronic hypoxia increases the expression of VEGF and CXCL12 in a cancer cell, factors that act on endothelial cells. VEGF increases the expression of CXCR4 and thus enhances the sensitivity of these cells to CXCL12. In turn, CXCL12 causes an increase in VEGF expression in endothelial cells. In addition, chronic hypoxia itself can act directly on endothelial cells. It increases the expression of CXCR4, CXCL12 and VEGF, which act in an autocrine manner on these cells. As a result of the described factors, angiogenesis occurs.

**Figure 5 ijms-22-00843-f005:**
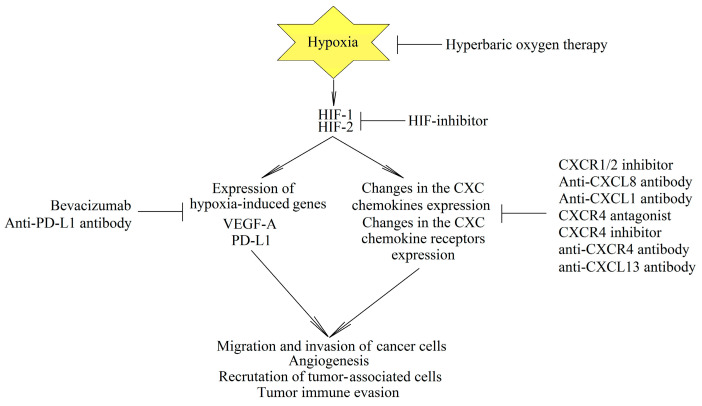
Hypoxia as an essential element of cancer mechanisms and as a therapeutic target. Hypoxia leads to an increase in the expression of genes essential for tumor progression, especially CXC chemokines, CXC chemokine receptors, and other genes encoding VEGF-A and programmed death-ligand 1 (PD-L1). As hypoxia occurs mainly in the tumor, cancer-specific therapy can be developed by using hyperbaric oxygen therapy and HIF inhibitors. Researchers are also investigating the potential of inactivating specific proteins whose expression is increased by hypoxia.

**Table 1 ijms-22-00843-t001:** Human CXC chemokines and their effect on selected cancer processes.

Official Name	Alternative Name	Receptor	Effect on the Recruitment and Accumulation of Cells into the Tumor Niche	Effect on Angiogenesis in a Tumor
CXCL1	GRO-α	CXCR2	MDSC, MSC, TAN, T_reg_	Angiogenic
CXCL2	GRO-β	CXCR2	MDSC, TAN	Angiogenic
CXCL3	GRO-γ	CXCR2	TAN	Angiogenic
CXCL4	PF-4	CXCR3	TIL, T_reg_	Angiostatic, lymphangiostatic
CXCL5	ENA-78	CXCR2	MDSC, TAN	Angiogenic
CXCL6	GCP-2	CXCR1, CXCR2	TAN	Angiogenic
CXCL7	NAP-2	CXCR2	TAN, TAM	Angiogenic
CXCL8	IL-8	CXCR1, CXCR2	MSC, TAM, TAN, MDSC	Angiogenic
CXCL9	MIG	CXCR3	TIL, T_reg_	Angiostatic
CXCL10	IP-10	CXCR3	TIL, T_reg_	Angiostatic
CXCL11	I-TAC	CXCR3, CXCR7	TIL, T_reg_	Angiostatic
CXCL12	SDF-1	CXCR4, CXCR7	MDSC, MSC, TAM, TAN, T_reg_	Angiogenic, lymphangiogenic
CXCL13	BCA-1	CXCR3, CXCR5	MDSC, T_reg_	Angiostatic
CXCL14	-	Unknown	CAF, TIL	Angiostatic
CXCL16	-	CXCR6, mCXCL16	MSC, TAM, TIL, T_reg_	Angiogenic
CXCL17	VCC-1	CXCR8	MDSC	Angiogenic

CAFs—cancer-associated fibroblasts; mCXCL16—membrane form CXCL16; MDSCs—myeloid-derived suppressor cells; MSCs—mesenchymal stem cells; TAMs—tumor-associated macrophages; TANs—tumor-associated neutrophils; T_regs_—regulatory T cells; TILs-tumor-infiltrating lymphocytes.

## Data Availability

Not applicable.

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
