# Peer review of "The Effect of Hypoxia on the Expression of CXC Chemokines and CXC Chemokine Receptors—A Review of Literature"

_ijms, 2021, doi:10.3390/ijms22020843_

Round 1
Reviewer 1 Report
The manuscript “The effect of hypoxia on the expression of CXC chemokines and CXC chemokine receptors – a review of literature” discuss the effect of hypoxia-induced changes on CXC chemokine/receptor expression. In addition, the review summarizes data on the use of drugs targeting the CXC chemokine system in cancer therapies. Overall the manuscript is clearly written and structured well. However, there were several comments with the information that has been presented.
Major points
- Although the authors listed a lot of evidence for the effects of hypoxia on the expression of various CXC chemokines and CXC receptors, the mechanisms of action for these effects are almost not mentioned, except for CXCL8. The authors are recommended to discuss more about the mechanisms to induce the chemokine expression by hypoxia.
- In addition, both hypoxia and CXC chemokines/receptors have multiple effects on the cancer progression such as immune cell attraction, cell proliferation, angiogenesis, and metastasis. It is necessary to explore the signaling pathway related to hypoxia, CXC chemokines and receptors expression and the consequence of the pathway regarding to cancer progression, instead of only discussing about the chemokine expression. Are there any investigated cancer therapy targeting the signaling pathway?
- HIF-1α can be induced by both oxygen-dependent and independent mechanisms. It may be meaningful to discuss whether the oxygen independent pathways inducing HIF-1α have any effect on CXC chemokines/receptors expressions.
Minor points
- The figure 1 is quite difficult to understand the meaning. Please improve the figure 1 and add more information into the figure legend.
- Please add condition to figure 2 (normoxia/hypoxia). The addition of “moderate hypoxia” condition into the figure 2 but not in the main text induces the confusion. Please describe more in the text.
- The titles of sections “ Ligands of receptors CXCR1 and CXCR2” and “5. CXCR3 receptor ligands” are appropriate. Please retitle.
- Some more tables and figures should be included to summarize the information in the Review article.
Author Response
The manuscript “The effect of hypoxia on the expression of CXC chemokines and CXC chemokine receptors – a review of literature” discuss the effect of hypoxia-induced changes on CXC chemokine/receptor expression. In addition, the review summarizes data on the use of drugs targeting the CXC chemokine system in cancer therapies. Overall the manuscript is clearly written and structured well. However, there were several comments with the information that has been presented.
Major points
- Although the authors listed a lot of evidence for the effects of hypoxia on the expression of various CXC chemokines and CXC receptors, the mechanisms of action for these effects are almost not mentioned, except for CXCL8. The authors are recommended to discuss more about the mechanisms to induce the chemokine expression by hypoxia.
While writing this paper we collected all available data on mechanisms of hypoxia influence for CXC chemokines. However, many mechanisms and effects of hypoxia on some CXC chemokines have not been studied. For this reason, we were not and are not able to provide this information. We hope that some readers will want to fill this gap by performing appropriate experiments in their laboratories.
- In addition, both hypoxia and CXC chemokines/receptors have multiple effects on the cancer progression such as immune cell attraction, cell proliferation, angiogenesis, and metastasis. It is necessary to explore the signaling pathway related to hypoxia, CXC chemokines and receptors expression and the consequence of the pathway regarding to cancer progression, instead of only discussing about the chemokine expression. Are there any investigated cancer therapy targeting the signaling pathway?
We have added a fragment about the importance of hypoxia and HIF in tumor progression, as well as a fragment on anticancer therapy directed against signaling pathway associated with hypoxia. However, intracellular signaling pathways activated by CXC chemokine receptors are not targeted in therapies. It is easier and more efficient to activate either the CXC chemokine receptor itself or the chemokine itself.
- HIF-1α can be induced by both oxygen-dependent and independent mechanisms. It may be meaningful to discuss whether the oxygen independent pathways inducing HIF-1α have any effect on CXC chemokines/receptors expressions.
Oxygen-independent pathways of HIF-1 activation are now included.
Minor points
- The figure 1 is quite difficult to understand the meaning. Please improve the figure 1 and add more information into the figure legend.
The Figure 1 has been corrected.
- Please add condition to figure 2 (normoxia/hypoxia). The addition of “moderate hypoxia” condition into the figure 2 but not in the main text induces the confusion. Please describe more in the text.
It has been corrected according to the Reviewer's suggestions.
- The titles of sections “ Ligands of receptors CXCR1 and CXCR2” and “5. CXCR3 receptor ligands” are appropriate. Please retitle.
It has been corrected.
- Some more tables and figures should be included to summarize the information in the Review article.
Additional 2 figures have been added.

Reviewer 2 Report
This manuscript does not fit the criterior of this journal.
At first, there is no effort to meet the format of IJMS.
It seems to be testing but not for publishing.
Please see the other reviews.
Author Response
This manuscript does not fit the criterior of this journal.
At first, there is no effort to meet the format of IJMS.
It seems to be testing but not for publishing.
Please see the other reviews.
The template used for formatting articles to be sent to IJMS was changed last month. Our manuscript has been re-formatted according to the current IJMS editorial guidelines.

Reviewer 3 Report
Line 74 This is a bit broadly spoken – HIF will also be activated in cycling hypoxia without ROS due to the transient reduction in degradation
My comments are as follows:
Line 81 it needs to be mentioned that PHD2/3 are hypoxia-inducible HIF targets – therefore a balance will be set between hypoxia-reduced activity and Hypoxia-induced abundance of PHDs
Line 125 affect
Line 183 Again, this is a bit broad – no HRE means (probably) not HIF-regulated
Line 195 was HIF-2 tested in ref 123, 124, 117 and 111?
Line 209 via HIF?
Line 268 Were TPH cells really differentiated into macrophages?
On several occasion you mention that both HIF-1 and NFκB activate gene expression, e.g. of CXCL10, through common activation of the gene promoter. IS there any model that would explain such a cooperative effect? Can you distinguish the effects of continuous vs. cycling hypoxia by the activation of these trx factors?
Line 347 CoCl2 is no longer regarded as a hypoxia mimetic – it has so many redox effects…
Line 486 Didn´t you want to leave out CXCL15?
Author Response
Rev3.
Line 74 This is a bit broadly spoken – HIF will also be activated in cycling hypoxia without ROS due to the transient reduction in degradation
We gave only a simplified model with the most important mechanisms. There are studies which show that during cycling hypoxia HIF-1a is phosphorylated by PKA, PKC, mTOR, which increases the stability and synthesis of this HIF subunit. However, the activation of these pathways is dependent on ROS.
My comments are as follows:
Line 81 it needs to be mentioned that PHD2/3 are hypoxia-inducible HIF targets – therefore a balance will be set between hypoxia-reduced activity and Hypoxia-induced abundance of PHDs
The information has been added according to the Reviewer's suggestions.
Line 125 affect
It has been corrected according to the Reviewer's suggestions.
Line 183 Again, this is a bit broad – no HRE means (probably) not HIF-regulated
Studies quoted in this paragraph show that HIF-1 affects the expression of CXCL3. We added information that this effect may be indirect due to lack of HRE in the CXCL3 gene promoter.
Line 195 was HIF-2 tested in ref 123, 124, 117 and 111?
HIF-2 was not studied in those papers. It is possible that in the quoted papers a model was used in which HIF-2 was not activated (but HIF-1 was) and CXCL5 expression depends only on HIF-2. We added this thought.
Line 209 via HIF?
The quoted article does not examine the significance of HIF in the influence of cycling hypoxia. Therefore, we did not give the mechanism.
Line 268 Were TPH cells really differentiated into macrophages?
In the methodology of the quoted work, the scientists differentiated THP-1 monocytes into macrophages using PMA and then polarized them using different cytokines. This information is now included in the text.
Delprat V, Tellier C, Demazy C, Raes M, Feron O, Michiels C. Cycling hypoxia promotes a pro-inflammatory phenotype in macrophages via JNK/p65 signaling pathway. Sci Rep. 2020 Jan 21;10(1):882.
On several occasion you mention that both HIF-1 and NFκB activate gene expression, e.g. of CXCL10, through common activation of the gene promoter. IS there any model that would explain such a cooperative effect? Can you distinguish the effects of continuous vs. cycling hypoxia by the activation of these trx factors?
In the section on hypoxia we wrote that chronic hypoxia reduces PHD activity, which activates both HIF and NF-κB pathway. NF-κB increases HIF-1alpha expression and thus leads to full activation of HIF during hypoxia. However, some other genes are also expressed. That is, in chronic hypoxia the expression of genes may depend on the NF-κB both directly and indirectly (via HIF-1a). In cycling hypoxia we have ROS which activate the HIF and NF-κB pathways. Therefore, NF-κB directly increases gene expression, but the indirect effect dependent on NF-κB =>HIF-1a is much smaller than in chronic hypoxia. This idea is highlighted in the current version of our review.
Line 347 CoCl2 is no longer regarded as a hypoxia mimetic – it has so many redox effects…
We are aware that CoCl2 is a chemical compound mimicking hypoxia and the results obtained from the use of this compound may not coincide with the results from the use of reduced oxygen levels. Even the cited paper by Muñoz-Sánchez et al. pointed out some differences between hypoxia and CoCl2 in the last sections. Therefore, we tried not to quote works where only this compound was used. Nevertheless, there were no works showing the influence of hypoxia on CXCR3 expression. We found only the work in which CoCl2 was used, so we quoted it with a note that these results may only indicate that hypoxia has the same effect on the expression of the described protein.
Muñoz-Sánchez J, Chánez-Cárdenas ME. The use of cobalt chloride as a chemical hypoxia model. J Appl Toxicol. 2019 Apr;39(4):556-570. doi: 10.1002/jat.3749.
Line 486 Didn´t you want to leave out CXCL15?
After reading the title, the reader will want to know the influence of hypoxia on chemokines CXCL1-17. Of course, we may omit CXCL15, but this will be inconsistent with the chosen purpose of discussing the entire group of CXC chemokines.

Reviewer 4 Report
Very well written review. Nicely described and informative schemes/tables are an additional value. The topic of the review – CXC chomokines in the context of cancers is very interesting and this review will certainly find many readers.
Author Response
Rev. 4
Very well written review. Nicely described and informative schemes/tables are an additional value. The topic of the review – CXC chomokines in the context of cancers is very interesting and this review will certainly find many readers.
We are very grateful for this review, thank you.

Round 2
Reviewer 1 Report
The revised manuscript was much improved.
Please change the titles as following:
- "Ligands of receptor CXCR3" to "Effects of hypoxia on the expression of CXCR3 ligands"
- "Ligands of receptors CXCR1 and CXCR2" to "Effects of hypoxia on the expression of CXCR1 and CXCR2 ligands"
Reviewer 2 Report
Authors still are not careful about the format of IJMS format of review.
Please read the IJMS guide for review and review exmaple. For example, there is no 1 - 12 in other review. There is still no conclusion.
Please merge of short section.